# Circadian Gene *PER2* Silencing Downregulates *PPARG* and *SREBF1* and Suppresses Lipid Synthesis in Bovine Mammary Epithelial Cells

**DOI:** 10.3390/biology10121226

**Published:** 2021-11-24

**Authors:** Yujia Jing, Yifei Chen, Shan Wang, Jialiang Ouyang, Liangyu Hu, Qingyong Yang, Mengzhi Wang, Bin Zhang, Juan J. Loor

**Affiliations:** 1State Key Laboratory of Sheep Genetic Improvement and Healthy Production, Xinjiang Academy of Agricultural Reclamation Sciences, Shihezi 832000, China; 2018205029@njau.edu.cn (Y.J.); yqy@mail.hzau.edu.cn (Q.Y.); 2College of Animal Science and Technology, Yangzhou University, Yangzhou 225009, China; cyfjaml@gmail.com (Y.C.); sunnyshan5233@163.com (S.W.); jialiangouyangyz@126.com (J.O.); liangyu.hu@wur.nl (L.H.); 3Department of Animal Sciences and Division of Nutritional Sciences, University of Illinois, Urbana, IL 61801, USA

**Keywords:** clock signaling, milk fat, lactation, transcription

## Abstract

**Simple Summary:**

The present study was constructed to determine the effects of the core circadian clock gene, Period 2 (*PER2*), on lipid synthesis in bovine mammary epithelial cells (BMECs). Data revealed that *PER2*-regulated genes were involved in fatty acid de novo synthesis, desaturation, TAG accumulation, and lipid droplet secretion in primary BMECs, partly by inhibiting *PPARG* and *SREBF1*. Our overall data suggests that *PER2* in bovine mammary cells plays a role in regulating milk fat synthesis directly, or via the activation of the transcription regulators *PPARG* and *SREBF1*. This study provides molecular evidence underscoring a link between the circadian clock and lipid metabolism in bovines.

**Abstract:**

*PER2,* a circadian clock gene, is associated with mammary gland development and lipid synthesis in rodents, partly via regulating peroxisome proliferator-activated receptor gamma (*PPARG*). Whether such a type of molecular link existed in bovines was unclear. We hypothesized that *PER2* was associated with lipid metabolism and regulated cell cycles and apoptosis in bovine mammary epithelial cells (BMECs). To test this hypothesis, BMECs isolated from three mid-lactation (average 110 d postpartum) cows were used. The transient transfection of small interfering RNA (siRNA) was used to inhibit *PER2* transcription in primary BMECs. The silencing of *PER2* led to lower concentrations of cellular lipid droplets and triacylglycerol along with the downregulation of lipogenic-related genes such as *ACACA*, *FASN*, *LPIN1,* and *SCD,* suggesting an overall inhibition of lipogenesis and desaturation. The downregulation of *PPARG* and *SREBF1* in response to *PER2* silencing underscored the importance of circadian clock signaling and the transcriptional regulation of lipogenesis. Although the proliferation of BMECs was not influenced by *PER2* silencing, the number of cells in the G2/GM phase was upregulated. *PER2* silencing did not affect cell apoptosis. Overall, the data provided evidence that *PER2* participated in the coordination of mammary lipid metabolism and was potentially a component of the control of lipid droplets and TAG synthesis in ruminant mammary cells. The present data suggested that such an effect could occur through direct effects on transcriptional regulators.

## 1. Introduction

In non-ruminants, peripheral clocks exhibit a rhythmic oscillation with autonomic rhythms in almost all peripheral tissues and organs, including the mammary gland [1,2,3]. In the bovine mammary gland, there is also evidence that a number of circadian rhythm genes, some of which have well-known metabolic roles, play a key role in mammary gland function [4,5].

In cows, indirect evidence indicates that endogenous circadian clock systems could represent important control points of the biological rhythms associated with physiology and biochemistry. For example, rectal temperatures, respiratory rates, plasma concentrations of metabolites (glucose, urea, and cholesterol) [6], and concentrations of the growth hormones [7] prolactin [8] and cortisone [9] exhibit circadian oscillations in lactating cows. The synthesis of milk components also appears to follow a rhythmic pattern, which can be regulated, in part, by a circadian mechanism [10,11,12].

In rodents, Period circadian regulator 2 (*PER2*), a core circadian clock gene, plays a role in lipid metabolism. *PER1/2*-deficient mice have reduced liver TG levels [13], suggesting that *PER* could be involved in lipid metabolism. Another study in a rat model revealed that *PER2* controls lipid metabolism via targeting *PPARG* expression [14]. Thus, *PER2* might represent a novel control point in the bovine mammary lipid metabolism. The *PER2* gene is not only associated with lipid metabolism, but it also shares inextricable molecular links with cell cycle oscillators [15,16]. In non-ruminants, the synchronized physiological signals of the circadian clock could lead to cell aggregation and division [17], indicating that the cell cycle is synchronized by the circadian clock. Although data supports an important role for *PER2* in lipid metabolism, such effects, and the molecular mechanisms in bovine mammary epithelial cells (BMECs), are largely unknown.

In the present study, we hypothesized that *PER2* regulated lipid metabolism and was potentially associated with cell cycle control in BMECs. To test this hypothesis, small interfering RNA (siRNA) was used to silence *PER2* expression in primary BMECs. We investigated the role of *PER2* on cell cycle activity and lipid synthesis at a transcription level, aiming to provide molecular evidence to support the link between the circadian clock and lipid metabolism in the bovine. 

## 2. Materials and Methods

### 2.1. The Isolation and Culture of Primary BMECs

The use of animals and the experimental procedures were approved by the Ethical Committee of Yangzhou University, Jiangsu Province, China (Approval Code: SYXK (Su)2021-0026). Three mid-lactation (average 110 ± 5 d in milk (DIM) and 34.6 ± 0.5 kg/d of milk) cows from the Yangzhou University dairy farm were used. Bovine mammary tissue was obtained using a published mammary biopsy method [18] and was washed thoroughly with PBS, containing 100 IU/mL of penicillin/streptomycin (Sigma-Aldrich, St. Louis, MO, USA). Primary BMECs were harvested by isolation and purification using 0.25% collagenase (Gibco, Grand Island, NY, USA) digestion as described by Hu et al. (2017) [19]. Cells were cultured in DMEM-F12 (Dubecco’s Modified Eagle Medium Nutrient Mixture F-12 (Ham), Gibco, Grand Island, NY, USA) containing 10% fetal bovine serum (Gibco, Grand Island, NY, USA), 50 IU/mL of prolactin (Gibco, Grand Island, NY, USA), 1 μg/mL of cortisol (Gibco, Grand Island, NY, USA), 0.5 μL/mL of Insulin–Transferrin–Selenium (ITS-G; Gibco, Grand Island, NY, USA), 10 ng/mL of the Epidermal Growth Factor (EGF; Peprotech, Rocky Hill, NJ, USA), 100 IU/mL of penicillin, 0.2 mg/mL of streptomycin (Sigma-Aldrich, St. Louis, MO, USA), and 2.5 ug/mL of amphotericin B (Solarbio, Beijing, China). Cells were routinely cultured at 37 °C with 5% CO_2_.

### 2.2. Cell Transfection

Small interfering RNA (siRNA) was used to inhibit *PER2* expression in primary BMECs. The primary BMECs were transfected with three siRNAs, targeting the *PER2* gene at 0, 12, 24, 36, 48, and 60 h, to screen for optimal siRNA and its transfection time point. The real-time PCR was employed to determine the transfection efficiency.

The primary BMECs were divided into a *PER2* silencing group and a non-transfected group (without transfection, blank control). Negative siRNA was used as negative control during the real-time PCR. SiRNA and negative siRNA was synthesized by GenePharma Co., Ltd., (Genepharma Co., Ltd, Shanghai, China) and the sequences are listed in Appendix A. The transfection protocol was as follows: primary BMECs were inoculated onto 24-well plates until reaching 70–80% confluence. Subsequently, the siRNA was transfected using a Transfection Reagent Kit (ViewSold Biotech, Beijing, China) according to the manufacturer’s protocol.

### 2.3. RNA Extraction and RT-PCR

Total RNA was extracted from BMECs with the TRIzol reagent (TIANGEN, Beijing, China) and was quantified by spectrophotometry at 260 nm using a NanoDrop 1000 spectrophotometer (NanoDrop Technologies, Wilmington, DE, USA). First-strand cDNA was synthesized with the FastQuant RT Kit (No. KR106, TIANGEN, Beijing, China) according to the manufacturer’s instructions. The primer sequences for the target and internal reference genes were designed with Oligo 6 (Appendix A) and were synthetized by Invitrogen Biotechnology Co., Ltd. (Invitrogen Biotechnology Co., Ltd, Shanghai, China).

The RT-PCR was conducted with the SuperReal PreMix Plus (SYBR Green, No. FP215) from TIANGEN Biotech Co., Ltd. (TIANGEN Biotech Co., Ltd, Beijing, China) in an Applied Biosystems 7500 Real-Time PCR System (USA) to determine the relative expression of the target genes. Reactions were as follows: 15 min at 95 °C, followed by 10 s at 95 °C, and then 32 s at 60 °C, for 40 cycles. Each sample was run in triplicate. The relative gene expression was calculated with the 2^−ΔΔCt^ method.

### 2.4. The Cell Proliferation Activity, Cell Cycles, and Apoptosis Assays

The primary BMECs on 24-well plates were exposed to *PER2* siRNA and the cell proliferation activity at 12, 24, and 36 h-transfections determined using the Cell Counting Kit-8 (CCK8, Dojindo, Japan) according to the manufacturer’s protocols. After 12, 24, and 36 h-transfection, the culture medium was aspirated and the cells were rinsed with PBS and cultured with DMEM-F12, containing a 10% CCK8, for 1.5 h. A 200 μL-culture medium was transferred to 96-well plates to determine absorbance at 450 nm with an MD-SpectraMax M5 plate reader (Molecular Devices Corporation, Sunnyvale, CA, USA). 

The cell cycle was assessed with the Cell Cycle and Apoptosis Analysis Kit (C1062, Beyotime, Shanghai, China). The cellular DNA content was quantified via flow cytometry (FACS LSRFortessa, BD, USA) at a 488 nm wavelength and was analyzed with the FlowJo software. The proliferation index (PI) was calculated as: PI = (S + G2/M)/(G0/G1 + S + G2/M) × 100%. Apoptosis was monitored by FITC-conjugated antibodies with Annexin V, using the Annexin V-FITC Apoptosis Detection Kit (C1062, Beyotime, Shanghai, China). Apoptotic cell numbers were determined by FACS LSRFortessa (BD, USA) as described in the manufacturer’s protocols. The expression of the apoptosis-related genes caspase-3, caspase-8, and P53 was determined by the real-time PCR and the primers listed in Appendix A.

### 2.5. Triacylglycerol Content and Lipid Droplet Determination

The lipid droplet content was determined with red oil O (Sigma, USA). Briefly, transfected cells were fixed in formaldehyde at 4 °C for 1 h and stained with 0.5% oil red O overnight at room temperature. Subsequently, samples were washed with 60% isopropyl alcohol several times. The lipid droplet content was quantified via counting under an inverted microscope (Olympus, Tokyo, Japan, CKX41). To quantify the lipid droplet density, each oil red O optical density (OD) value at 500 nm was measured, after de-staining with isopropanol for 30 min, using a MD-SpectraMax M5 plate reader (Molecular Devices Corporation, CA, USA).

The triacylglycerol (TAG) content in primary BMECs was determined with the Tissue Triglyceride Assay Kit (Applygen, Beijing, China) following the manufacturer’s instructions. The specific steps were as follows: cells were lysed and centrifuged at 2000× *g* for 5 min to harvest the supernatant. The supernatant was incubated at 70 °C for 10 min and was then collected for an enzyme activity measurement. The samples, and a 4 mM glycerol standard at different dilutions, were measured at 550 nm with a microplate reader. The TAG content in samples was calculated using a standard curve, following the manufacturer’s protocols.

To investigate lipogenesis at the transcription level, the gene expressions of *PPARG, SREBF1, mTOR, ACACA, FAS, LPIN1, SCD,* and *LPL* were determined using the real-time PCR. Primers for these genes are listed in Appendix A.

### 2.6. Statistical Analyses

SPSS software version 22.0 was used for statistical analyses. Each experiment was performed in triplicate on different days with three experimental replicates at each time. Differences in cell cycles, cell apoptosis, and lipid synthesis between the *PER2* silenced and non-transfected groups were analyzed via a Student’s *t* test. The differences in the target gene expression between *PER2* silencing, the negative control, and the non-transfected group were evaluated by a one-way ANOVA. Statistical significance was declared at *p* < 0.05.

## 3. Results

### 3.1. Screening for Transfection Efficiency

The BMECs transfected with siRNA-*PER2*-b for 36 h had the lowest *PER2* mRNA abundance, with an 84.7% inhibitory effect compared to normal BMECs (0 h) (Figure 1). Thus, all subsequent tests involving *PER2* silencing were performed using BMECs transfected with siRNA-*PER2*-b for 36 h.

### 3.2. The Cell Proliferation Activity and Cell Cycles

The cell proliferation activity did not differ after the transfection of *PER2*-siRNA for 12, 24, and 36 h (*p* > 0.05) (Figure 2A). Based on the cell cycle analysis (Figure 2B,C), *PER2* silencing increased cell numbers in the G2/M phase (*p* < 0.05), but there was no significant effect on cells in the G0/G1 and S phases (*p* > 0.05). The proliferation index (PI) was higher in the *PER2* silence group (43.501%) than the non-transfected group (34.282%), but there was no significant difference (*p* > 0.05).

### 3.3. Cell Apoptosis

The cell apoptosis index (the proportion of apoptotic cells among the total counted cells) was not affected by *PER2* silencing (*p* > 0.05) (Figure 3A). Among the genes measured (Figure 3B), caspase-8 mRNA expression was upregulated by *PER2* silencing (*p* < 0.05), but its downstream gene, caspase-3, did not differ between the groups (*p* > 0.05). This indicated that *PER2* silencing might have upregulated caspase-8 but failed to drive caspase-3 expression, hence preventing an increase in apoptosis.

### 3.4. Lipid Synthesis

The concentrations of triacylglycerol, lipid droplets, and lipogenic gene expressions were measured as indicators of lipid sythesis. Figure 4A depicts lipid droplets in BMECs by oil red O staining. In comparison with the non-transfected group, *PER2* silencing led to a 29.8% decrease in lipid droplet density and 38.86% decrease in lipid droplet quantity (*p* < 0.05) (Figure 4B,C). Furthermore, the cellular triacylglycerol concentration (*p* < 0.05) significantly decreased by 48.93% after *PER2* silencing (Figure 4D). At the transcription level, the expression of the lipid transcription regulators *PPARG* and *SREBF1* was downregulated after *PER2* silencing in comparison with the non-transfected group (*p* < 0.05). The expression of the lipid synthesis-related genes (Figure 5) *mTOR, ACACA, FAS, LPIN1,* and *SCD* was downregulated by *PER2* silencing (*p* < 0.05), while *LPL* mRNA expression was not affected (*p* > 0.05).

## 4. Discussion

The milk fat content in dairy cows follows a rhythmic pattern and is thought to be regulated by circadian rhythms. The present study was an investigation to better understand the role of the clock gene *PER2* on milk fat synthesis gene networks in BMECs. We employed *PER2* silencing experiments in BMECs and observed that *PER2* silencing led to a significant inhibition of lipid synthesis-related gene expression. A *PER2* silencing model was proposed based on the results from the present study (Figure 6). The findings suggest that *PER2* participates in the coordination of mammary lipid metabolism and may be a component of the control of lipid synthesis in ruminant mammary cells. Thus, these data provide an important reference to molecular links between the circadian clock and lipid metabolism in bovine mammary cells. 

### 4.1. PER2 Silencing Downregulates Mammary Lipogenic Genes

*PPARG* and *SREBF1* play a central role in lipid metabolism. In this study, the downregulation of the lipid-related transcription regulators *PPARG* and *SREBF1* in BMECs after *PER2* silencing underscored the potential for *PER2* to participate in the regulation of mammary lipid synthesis. Consistent with our findings, previous studies revealed that circadian clock genes control lipid metabolism in non-ruminants [20,21,22]. *PER2* could directly control *PPARG* expression as demonstrated by Grimaldi et al. [14] where the knockout of *PER2* inhibits lipid metabolism by directly downregulating *PPARG* in mice. Furthermore, a concerted function between *SREBF1* and *PPARG* was reported in regulating lipid synthesis in ruminants [23]. Thus, by inhibiting *PPARG* and *SREBF1* in BMECs, *PER2* could alter various aspects of lipid synthesis in BMECs.

A recent study from our laboratory (not published) aimed to determine circadian rhythms in mRNA expressions of *PER2* and lipogenic genes in normal BMECs. We found the expression of both *PER2* and *PPARG* kept decreasing between 12 and 24 h after determination. This suggested the existence of a synchrony of *PER2* and *PPARG* expression in primary BMECs. We further analyzed the rhythms of *SREBF1* and observed that the decrease in *SREBF1* occurred approximately 4 h after the decrease of *PER2* and *PPARG*, with the decrease continuing between 16 and 24 h after analysis. Together, these findings are consistent with the present study, indicating a similar pattern of change with the circadian rhythm in *PER2*, *SREBF1,* and *PPARG* expression.

Data indicated, at least in non-ruminants, that the expression of *SREBF1* and *PPARG* is partly under the control of the mechanistic target of *mTOR* [24,25]. Non-ruminant studies revealed that the *mTOR* pathway could regulate lipid synthesis by integrating various molecular signals culminating in the activation of adipogenesis/lipogenesis through *SREBF1* and *PPARG* [26,27,28,29]. *mTOR* is essential for regulating *SREBF1* at both transcriptional and post-translational levels [30,31,32], and *mTORC2* can control lipid synthesis in BMECs by regulating the lipogenic gene expression through *PPARG* in ruminants [33]. In this study, the downregulation of *mTOR* at the mRNA level, as well as *SREBF1* and *PPARG*, indicated a strong mechanistic link, whereby *PER2* downregulated *SREBF1* and *PPARG* through *mTOR*.

### 4.2. PER2 Silencing Suppresses Cellular TAG Accumulation and Lipid Droplet Formation

The TAG and lipid droplet levels reflected the lipid synthesis capacity in BMECs. Lipid synthesis included the de novo synthesis of fatty acids and the esterification of newly synthesized fatty acids into TAG [34]. Lipid droplets in BMECs contained a cytosolic TAG-rich core, which is the end result of lipogenesis and esterification. The lower mRNA expression of TAG-related genes (*SCD, ACACA, LPIN1* and *FAS*) and lipid droplet secretion-related genes (*LPIN1*) when *PER2* was silenced represented a possible mechanism for the decrease in TAG and lipid droplet content. Thus, the fact that *PER2* silencing decreased lipid droplets and cellular TAG concentrations in BMECs is further evidence that this clock gene can regulate the basic mechanisms associated with lipid synthesis in BMECs.

The downregulation in the expression of *FAS* and *ACACA*, as well as *SCD*, when *PER2* was silenced, indicated that the inhibition of de novo fatty acid synthesis by *PER2* silencing contributed to the lower TAG content in BMECs. Both *PPARG* [35,36,37] and *SREBF1* [38,39,40] can regulate TAG synthesis. Our findings were consistent with previous studies where the inhibition of *SREBF1* activity was associated with the downregulation of *SCD, ACACA,* and *LPIN1* [25,41]. In accordance with the increased activity of *PPARG*, it led to the upregulation of *SCD* and *LPIN1* expression [35]. *LPIN1*, a target gene of *PPARG* in ruminants [23], is one key gene associated with lipid droplet formation. The downregulation of *LPIN1,* when *PER2* was silenced, agreed with the decrease in *PPARG* expression and the content of lipid droplets, indicating that *PER2* also exerted its effect on lipid droplet formation by regulating *LPIN1*. Collectively, the decrease in lipid droplets and cellular TAG concentrations in BMECs by *PER2* silencing was consistent with the mRNA expression data. Thus, this clock protein is associated with various aspects of lipid synthesis in BMECs.

### 4.3. PER2 Silencing Regulates the Cell Cycle, but Not Apoptosis

Cells operate in strict accordance with the G1-S-G2-M cycle and maintain normal biological rhythms under the control of the cell cycle network, Cyclins-CDKs-CKIs [17]. In this study, we observed that the proliferation index was not significantly influenced by *PER2* silencing. Furthermore, the increased number of cells in the G2/M phase suggested that *PER2* silencing can upregulate the DNA proliferation activity and cellular mitosis to induce more cells into the G2/M. Consistent with our findings, there is compelling evidence for a mechanistic link between *PER2* and the cell cycle [42,43], where the downregulation of *PER2* accelerated the growth of breast cancer cells [44]. Our results showed that *PER2* silencing did not affect cell apoptosis, which was consistent with Hu et al. [19]. Caspase-8, as the initiator of apoptosis at the top of the caspase signaling cascade [45], was significantly upregulated by *PER2* silencing, although its downstream gene caspase-3 was not significantly influenced. It is possible that the higher expression of caspase-8 failed to drive the upregulation of downstream genes in the caspase signaling pathway, hence explaining the lack of an effect on cell apoptosis when PER2 was silenced.

## 5. Conclusions

The clock protein *PER2* regulates genes involved in lipid metabolism, including fatty acid de novo synthesis, fatty acid desaturation, TAG accumulation, and lipid droplet secretion in primary BMECs. The inhibition of *PPARG* and *SREBF1* when *PER2* is silenced appears to be an important cause for the reduction in lipid synthesis. Our overall data suggests that *PER2* in bovine mammary cells plays a role in governing milk fat synthesis directly, or through the activation of the transcriptional regulators *PPARG* and *SREBF1*. This study provides molecular evidence in support of a link between circadian clocks and lipid metabolism in bovines.

## Figures and Tables

**Figure 1 biology-10-01226-f001:**
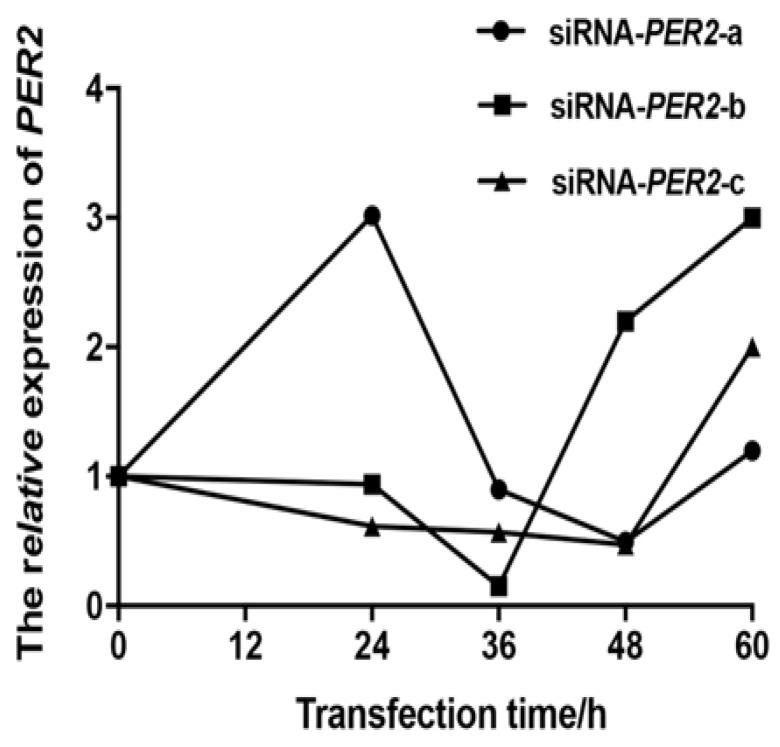
*PER2* gene mRNA expression of siRNA-*PER2*-a/b/c transfected primary BMECs were determined at 0, 12, 24, 36, 48, and 60 h by RT-PCR. Values are means ± SEM (n = 3).

**Figure 2 biology-10-01226-f002:**
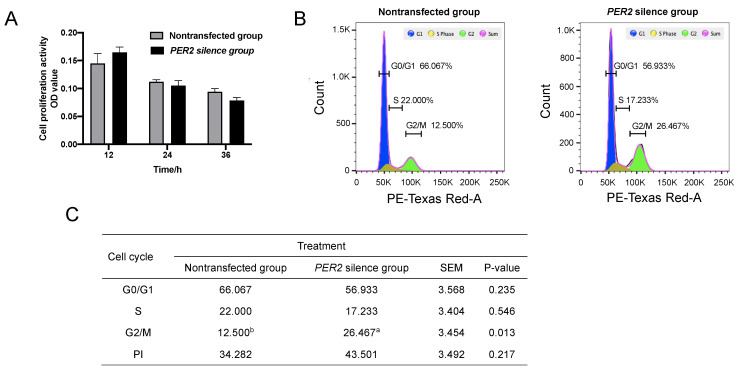
Cell cycles and proliferation in primary BMECs were determined after *PER2* silencing. Values are means ± SEM (n = 3). (**A**) The cell proliferation activity was determined at 12, 24, and 36 h after *PER2*-siRNA transfection using the CKK8 method. (**B**) The cell cycle was determined by flow cytometry and analyzed by FlowJo software after *PER2* silencing. (**C**) Cell cycle ratios (S/G2/M phase) were calculated based on result A. PI: proliferation index = (S + G2/M)/(G0/G1 + S + G2/M) × 100%. Different lowercase letters indicate values in three groups with significant difference (one-way ANOVA, *p* < 0.05).

**Figure 3 biology-10-01226-f003:**
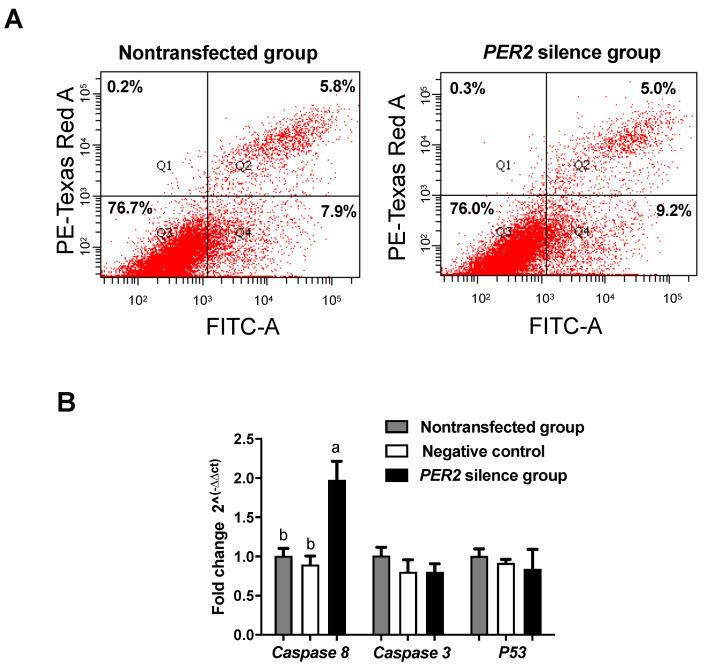
Cell apoptosis and apoptosis-related mRNA expressions were detected after *PER2* gene silencing in primary BMECs. Values are means ± SEM (n = 3). (**A**) Flow cytometry analysis of cell apoptosis was monitored by FITC-conjugated antibodies to Annexin V. Apoptotic cell numbers were determined by flow cytometry. (**B**) Apoptosis-related mRNA expression after *PER2* gene silencing. The BMECs transfected with negative siRNA were used as the negative control group. Relative mRNA expression was calculated with the 2^−ΔΔCt^ method. Significant differences in adhesion compared to the non-transfected group were found at *p* < 0.05 using one-way ANOVA. Different lowercase letters indicate values in three groups with significant difference (one-way ANOVA, *p* < 0.05).

**Figure 4 biology-10-01226-f004:**
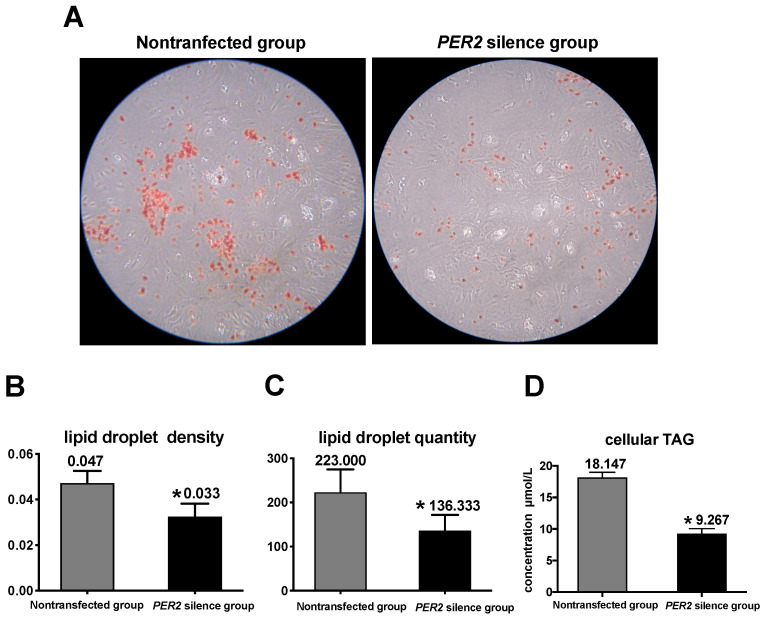
*PER2* silencing effects on lipid synthesis in primary BMECs. Values are means ± SEM (n = 3). (**A**) lipid droplet accumulation after *PER2* silencing. Lipid droplets were determined by oil red O staining and observed at 200× magnification under an inverted microscope. (**B**) Lipid droplet density. The density of the lipid droplet was determined by oil red O staining using a Microplate reader (510 nm). (**C**) Influence of *PER2* silencing on lipid droplet quantity. The lipid droplet quantity was obtained by counting under an inverted microscope. (**D**) Mean triglyceride levels (TAG) after *PER2* silencing were evaluated via a commercial TAG kit. Asterisks indicate statistically significant difference from nontransfected group (Student’s *t* test): * *p* < 0.05.

**Figure 5 biology-10-01226-f005:**
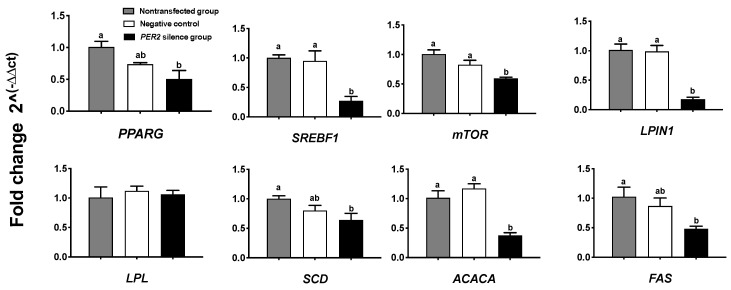
Effects of *PER2* silencing on the mRNA expression of genes associated with mammary lipid synthesis. The BMECs transfected with the negative siRNA were used as the negative control group. Relative mRNA expression of target genes was calculated with the 2^−ΔΔCt^ method. Values are means ± SEM (n = 3). Significant differences in expression compared with the non-transfected group was declared at *p* < 0.05 using one-way ANOVA. *PPARG*: peroxisome proliferator-activated receptor gamma. *SREBF1*: sterol regulatory element binding protein-1. *mTOR*: mammalian target of Rapamycin. *LPIN1:* Lipin 1. *LPL*: lipoprotein lipase. *SCD*: Stearoyl-CoA desaturase. *ACACA*: acetyl-CoA carboxylase alpha. *FAS*: fatty acid synthetase. Different lowercase letters indicate values in three groups with significant difference (one-way ANOVA, *p* < 0.05).

**Figure 6 biology-10-01226-f006:**
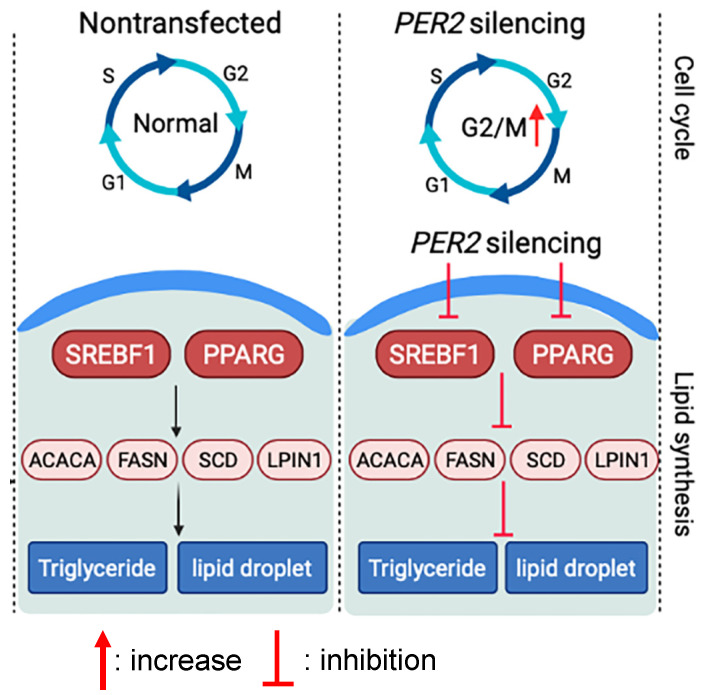
Proposed model of the effects of *PER2* silencing on lipid synthesis and cell cycle activity in primary bovine mammary epithelial cells based on results from the present study.

## Data Availability

Not applicable.

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
