# Peer review of "Circadian Gene *PER2* Silencing Downregulates *PPARG* and *SREBF1* and Suppresses Lipid Synthesis in Bovine Mammary Epithelial Cells"

_biology, 2021, doi:10.3390/biology10121226_

Round 1

Reviewer 1 Report

In the study titled with “Circadian gene PER2 silencing suppresses lipid synthesis in bovine mammary epithelial cells partly via inhibition of PPARG and SREBF1” by Lacey Yujia Jing et al. reported potential functions of PER2 in regulation of lipid metabolism in bovine mammary. The author performed PER2 siRNA knockdown and tested different processes/indexes and found that lipid synthesis was impaired and related regulator genes were downregulated. While there are still some weak points in the study and concerns in the manuscript. Unless the author could improve or modify all the weak points, then this manuscript could be considered for publication in a revised version.

1)There are extensive English grammar errors in the manuscript. It is clear the author does not have a strong grasp of the English language and this makes it hard to follow. The author needs to improve the overall writing of manuscript with either professional English language editing service or benefit from someone of native English-speaking areas.

2)The experiments from figure 1 and section 3.1 is not properly performed, and the data is not well presented. The author did not show the control PER2 expression at different timepoint. The current conclusion that the best knockdown point after 36 hours treatment is not persuasive. The author should perform the control experiments by using GFP-siRNA and normalize the expression level of PER2 to the control, then evaluate the best siRNA and optimal transfection time. The GFP-siRNA control should be applied in all the downstream experiments.

3)The author should add more information about the flow cytometry plot in figure 2B which will make the plot easier to digest. In figure 2C, as the P-value from the comparison of PI between control and siRNA group is much large than 0.05, which indicate that there is no difference. The author should not say there is an upward tendency, please correct the related narrations in the manuscript.

4)In figure 3, the author need to specify what is the negative control group showing in figure 3B(and also in figure 5). At the same time, the author should explain the possible reason of Caspase 8 upregulation in section 3.3 as the apoptosis index was not changed upon PER2 knockdown.

5)The barplots from figure 4B and 4C could be presented in a better way. I’d like to recommend the author to present the control group on the left and siRNA group on the right, otherwise it will confuse the reader when compared with oil red O staining in figure 4A.

6) In terms of the model in figure 6, I think the author should present it in an opposite way. Instead of PER2 silencing model, the author could present under normal/control conditions, how PER2 regulate downstream gene, i.e., SREBF1, PPARG which further regulate lipid synthesis and also involved in cell cycle.

7)In order to further confirm that PER2 regulates lipid metabolism through inhibition of PPARG and SREBF1, the author should perform the PPARG and SREBF1 knockdown to see whether they can observe the similar phenotype. And the author should perform the PPARG and SREBF1 overexpression along with PER2 knockdown to show the rescue effect of these downstream genes. These experiments will solidify the conclusions of this manuscript.

Minor Points:

1)Line13-14, “effects of core circadian. clock gene” should be “effects of core circadian. clock gene”.

2)The description about the siRNA and transfections from line 87-87 were repeated in line 93-95, the author should delete the repeated contents.

Reviewer 2 Report

Jing et al.,

Journal Biology (ISSN 2079-7737)

Manuscript ID biology-1416563

Circadian gene PER2 silencing suppresses lipid synthesis in bovine mammary epithelial cells partly via inhibition of PPARG and SREBF1.

Major (General or Overall) Comments to the Authors.

The study aims to test the hypothesis that PER2 exerts regulation on the cell cycle and might be associated with lipid metabolism in bovine mammary cells. The used primary bovine mammary epithelial cells (BMEC) to silence PER2 to attempt to quantify cell cycle activity and transcriptional control of lipogenesis. Overall, the manuscript is well written and methods seem appropriate to answer the questions about the effects of PER2 gene silencing on cell cycle and lipid transcription and TAG accumulation. Experimental design and statistics need to be better described. Specifically, replicates per treatment are not mentioned. Furthermore, I have some concerns with the statistical analyses. For example in Figure 5, the authors have 3 treatments but analyze using t-tests and not an ANOVA with a post hoc comparison to assign letter differences? This seems incorrect and needs to be addressed/justified and data likely need reanalyzing.  

Specific comments by line.

Ln 21-25: Please explain what PER2 is (i.e., circadian clock gene) in your 1st sentence so that the audience can understand the second and third sentences in this section.

Ln 47-48: What do you mean by stating that circadian clock systems could control biological rhythms associated with biochemistry? What do you mean by biochemistry in particular? Please clarify in text.

Ln 50: (glucose, urea, and cholesterol) 6. Is “6” a reference. Need to put in [ ].

Ln 50-60: Please expand on this idea to help readers better understand your hypothesis: “PER2 gene is not only connected with lipid metabolism, but also shares inextricable molecular links with cell cycle oscillators [15,16].. What does it mean to have links with cell cycle oscillators? In which ways would the cell cycle be influenced?

Ln 72-73: Protocol # available?

Ln 90: Why until 70-80% confluence?

Ln 120-121: what does this assay tell you, the stage of the cell cycle the cells are at?

Ln 142-148: please clearly define the number of replicates per assay/timepoint.

Ln 150-156: Was PER 2 being expressed at t 0? Your timeline is relative to t 0, but is not clear what the basal expression level was. Is time 0 the point of 70-80% confluence?

Ln 173-180: panel B white box. What is Nagtive? Negative? Please revise here and also make sur you mention your groups in the methods.   

Ln 190-195: 2 editsb needed here. 1) lipid droplet quality needs its own panel. 2) Please describe number of replicates/treatment of your experiments in the figures caption here (Fig 4) and in previous figures.

Ln 190: Can you really say “milk fat synthesis” if no milk is being produced? Is this not more “lipid accumulation/synthesis of primary BMCEs. Do primary cells make milk?

Figure 5: I am not convince these statistical analysis is appropriate. You hae 3 grous but did not run an ANOVA with posthoc test?

Ln 197-199: This caption is incomplete. Please add panel letters and descriptions for each of the genes  shown here. Each gene needs to be described (e.g., FAS: fatty acid synthase)

Ln 200: The illustration indicates the effects of PER2 silencing on cell cycle are independent from those on lipogenesis. Is that the case? Wouldn’t changes in the cell cycle impact lipogenesis as well? What does it mean to indicate that there is no effects on the cell cycle (dotted line) but that there is an increase in G2/M (red vertical arrow)?

Round 2

Reviewer 1 Report

The authors figured out most of the issues I concerned and made detailed responses to my comments. In view of the improved manuscript with much better quality, I suggest the publication of this work in this journal.

Reviewer 2 Report

I thank the authors for considering the comments and suggestions I made. The manuscript has clearly improved in terms of the quality of descriptions os experiemntal design and statistics. 

No further edits requested from my side.